# Can We Edit Multimodal Large Language Models?

**Siyuan Cheng**♣♥♡*, **Bozhong Tian**♣♥*, **Qingbin Liu**♡, **Xi Chen**♡†,
**Yongheng Wang**◇, **Huajun Chen**♣♥♠, **Ningyu Zhang**♣♥†

♣ Zhejiang University ♥ Zhejiang University - Ant Group Joint Laboratory of Knowledge Graph
♠ Donghai Laboratory ♡ Platform and Content Group, Tencent ◇ Zhejiang Laboratory
{sycheng,tbozhong,huajunsir,zhangningyu}@zju.edu.cn
{qingbinliu,jasonxchen}@tencent.com,wangyh@zhejianglab.com

## Abstract

In this paper, we focus on editing Multimodal Large Language Models (MLLMs). Compared to editing single-modal LLMs, multimodal model editing is more challenging, which demands a higher level of scrutiny and careful consideration in the editing process. To facilitate research in this area, we construct a new benchmark, dubbed **MMEdit**, for editing multimodal LLMs and establishing a suite of innovative metrics for evaluation. We conduct comprehensive experiments involving various model editing baselines and analyze the impact of editing different components for multimodal LLMs. Empirically, we notice that previous baselines can implement editing multimodal LLMs to some extent, but the effect is still barely satisfactory, indicating the potential difficulty of this task. We hope that our work can provide the NLP community with insights[1].

## 1 Introduction

With the widespread deployment of Large Language Models (LLMs) (Zhao et al., 2023), the necessity to maintain their knowledge accurate and current without incurring significant retraining costs is becoming increasingly paramount (Sinitsin et al., 2020). Previous research has introduced knowledge editing methodologies designed to incrementally infuse a language model with a new set of facts (Mitchell et al., 2022a; Han et al., 2023; Hartvigsen et al., 2022; Zhong et al., 2023; Gandikota et al., 2023; Yao et al., 2023).

Different from single-modal model editing, the task of editing multimodal LLMs presents considerable challenges, given their inherent diversity and complexity. Specifically, incorrect outputs from multimodal models may stem from the synergistic effects of various modalities. Incorrect outputs

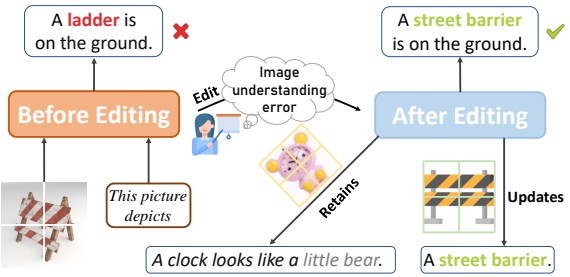

Figure 1: Overview of the **multimodal model editing** task. The editing target is to update the model's understanding of the edited input (e.g., image or text), while ensuring its interpretation of unrelated inputs remains as consistent as possible.

may stem not just from LLMs, analogous to human errors like misreading or misrecognition (e.g., color blindness affecting color identification in images). As shown in Figure 1, before the editing, the model misidentified the object as a "ladder" instead of the correct "barrier", resulting in an erroneous prediction. After the editing, the model accurately recognized the "barrier". Note that the utility of multimodal LLMs (Yin et al., 2023) is increasing, yet there is a lack of corresponding dataset resources and benchmarks for editing multimodal large language models.

To facilitate research in this area, we take the first step to construct a **M**ultimodal **M**odel **Edit**ing benchmark: dubbed as **MMEdit**, which encompass two sub-tasks: Editing VQA and Editing Image Captioning. Specifically, we follow single-modal model editing approaches (Mitchell et al., 2022a; Cao et al., 2021; Mitchell et al., 2022b) to construct the datasets, which extends the previous evaluation principle, namely **Reliability**[2], **Locality**[3], and **Generality**[4], to multimodal settings.

For Reliability evaluation, we start with rigorous data collection, gathering underperforming multi-

---

*Equal contribution.
†Corresponding author.
[1]Code and dataset are available in https://github.com/zjunlp/EasyEdit.

[2]The metric used to measure the success of editing target.
[3]It measures whether unrelated facts retain their outputs.
[4]It measures the success of editing related knowledge.

modal model data to create a dedicated reliability editing dataset (§3.2.1). For Locality evaluation, we split it into the textual and multimodal locality to evaluate the stability of multimodal LLMs (§3.2.2). For Generality evaluation, similar to Locality, we divide it into textual and multimodal generality and utilize ChatGLM (Du et al., 2022), and Stable Diffusion (Rombach et al., 2022) to generate rephrased text as well as rephrased images for evaluation (§3.2.3). We evaluate several knowledge editing approaches on **MMEdit**. Empirically, we notice that current editing approaches are effective for editing the textual model in the multimodal language model but not as effective for editing the vision module. For example, in editing the language module of the BLIP-2 model, the reliability of MEND can reach 92.6%, but only attain 14.1% if editing the vision module, indicating the potential difficulty and opportunities of this task. In general, our primary contributions are as follows:

- We take the first step to investigate editing multimodal LLMs, which extends model editing to multimodal settings.

- We propose **MMEdit**, a new benchmark, to evaluate the reliability, locality, and generality of multimodal model editing approaches.

- We conduct experiments with various baselines, demonstrating that while current methodologies can somewhat aid in multimodal editing, the outcomes still fall short of complete satisfaction. We will make the code and datasets publicly available for future research purposes.

## 2 Related Work

### 2.1 Multimodal Language Models

Multimodal Learning (MML) (Xu et al., 2022a; Yin et al., 2023) provides a holistic approach to crafting AI models that can extract and correlate information from various data modalities. Due to its societal significance, MML has established a foothold in the research community, solidifying itself as a crucial field of study over the past decade. Vision-language pre-training is one of the important branches of MML, which aims to learn multimodal foundation models with improved performance on various vision and language tasks. Vision Transformer (ViT) (Dosovitskiy et al., 2021) is a seminal work that contributes an end-to-end

solution by applying the encoder of Transformers to images. CLIP (Radford et al., 2021) proposes a method, which uses multimodal pre-training to convert classification as a retrieval task that enables the pre-trained models to tackle zero-shot recognition. Recently, the advancement of LLMs, such as LLaMA (Touvron et al., 2023), BLOOM (Scao et al., 2022), and ChatGPT (OpenAI, 2022), has been bolstered by scaled-up training data and increased parameters, yielding significant recent success. These models showcase impressive language understanding, generation, and knowledge reasoning capabilities, enhancing their ability to comprehend natural language and generate high-quality, context-based text. The evolution of large language models has spurred the widespread use of auto-regressive language models as decoders in vision-language tasks. Utilizing cross-modal transfer, this approach enables knowledge sharing between language and multimodal domains (Gao et al., 2023; Liu et al., 2023; Li et al., 2023a; Ye et al., 2023; Zhu et al., 2023; Li et al., 2023b; Zhang et al., 2023).

### 2.2 Model Editing

LLMs (Zhao et al., 2023) primarily derive knowledge from the training corpus. Yet, the quality of the dataset is not always guaranteed, potentially integrating harmful or incorrect information into the model (Hernandez et al., 2023). One solution is retraining models with updated knowledge, though this might be unfordable and difficult to implement. Alternatively, fine-tuning with a few updated facts could be considered, but it risks over-fitting and catastrophic forgetting (Zhai et al., 2023). To address these issues, (Sinitsin et al., 2020) proposes Model Editing, which aims to efficiently and accurately alter the factual knowledge stored within models. This approach is applied in various domains (Mao et al., 2023; Onoe et al., 2023; Xu et al., 2022b; Wang et al., 2023a; Li et al., 2023c; Cheng et al., 2023), with an increasing number of studies investigating the impact of editing (Ilharco et al., 2023; Gupta et al., 2023; Hase et al., 2023a; Cohen et al., 2023; Wu et al., 2023; Wang et al., 2023b; Gandikota et al., 2023; Li et al., 2023d; Hase et al., 2023b). Presently, there are three primary types of model editing approaches: 1) Meta-learning Method, 2) Locate-Then-Edit Method, and 3) In-Context Knowledge Editing Method.

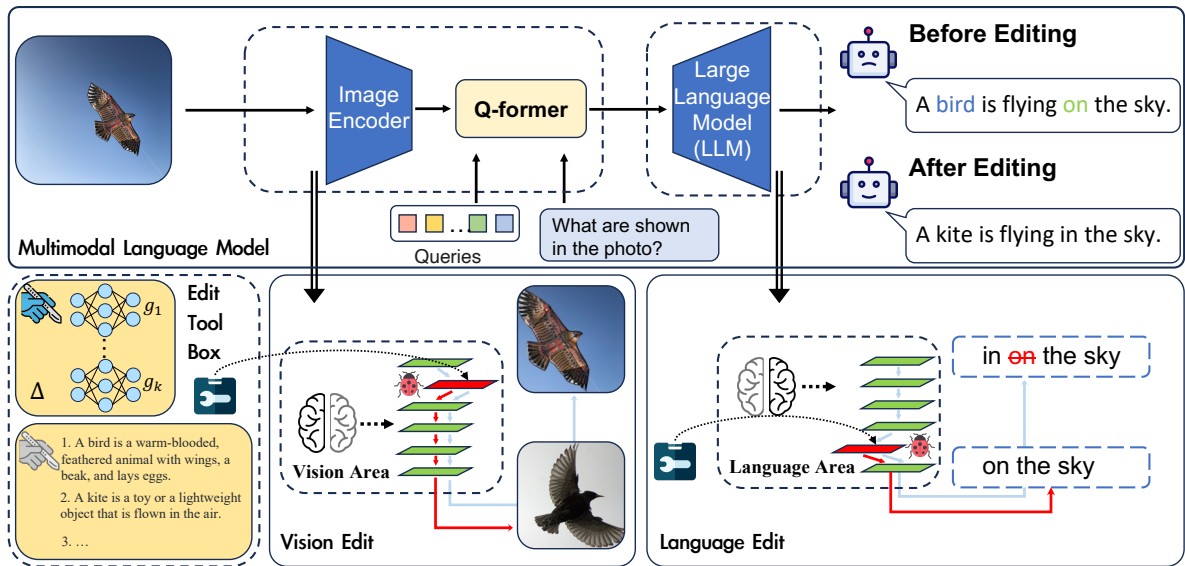

Figure 2: Utilizing multimodal LLM (e.g., BLIP-2 OPT) as an example, we dissect the comprehensive multimodal LLM into two components (Vision module and Textual module). The model's erroneous output could potentially stem from either or both of these modules. Drawing an analogy with human errors in "vision" and "speech", we apply model editing methods to these two components, thereby changing the model to refine its output.

**Meta-learning Method.** MEND (Mitchell et al., 2022a) and Knowledge Editor (KE) (Cao et al., 2021) propose approaches involving an external editor, capable of learning the optimal parameter set, $\theta$, for knowledge updating, while concurrently imposing constraints to maintain model stability. CaliNET (Dong et al., 2022) and T-Patcher (Huang et al., 2023), drawing inspiration from (Dai et al., 2022), introduce additional trainable parameters into the feed-forward module of Pretrained Language Models. SERAC (Mitchell et al., 2022b) utilize an explicit memory to store edits and learns to reason over them to modulate the base model's predictions as needed.

**Locate-Then-Edit Method.** ROME (Meng et al., 2022a) proposes approaches that employ causal mediation analysis to identify the area for editing. ROME discovers that memorized factual associations can be pinpointed to a specific location within a GPT model. However, a notable limitation of ROME is its ability only to edit one fact at a time. To address this, Meng et al. (2022b) proposes a new method known as MEMIT, which is a successor to the previous work ROME, which performs a rank-one modification of the MLP weights of a single layer to write a memory into the model directly.

**In-Context Knowledge Editing Method.** In-Context Learning (ICL) (Brown et al., 2020) signifies a training-free paradigm where knowledge is obtained from demonstrations directly concatenated within the input context. A novel editing paradigm has recently emerged that capitalizes on the ability of LLMs to comprehend context (Zheng et al., 2023), thereby enabling the performance of context-based model editing, guiding the model's generation process, and offering an efficient, lightweight approach to model editing.

Model editing methods to date largely cater to single-modal scenarios, leaving a gap in multimodal editing. To the best of our knowledge, we are the first to investigate multimodal model editing for LLMs and provide a new benchmark to facilitate research in this area.

## 3 Editing Multimodal LLMs

We illustrate the proposed task of multimodal editing in Figure 2. We will introduce the task definition (§3.1), dataset construction details in (§3.2), the multimodal models (§3.3), and the baselines (§3.4) we used in the experiments.

### 3.1 Task Definition

Assuming we have a multimodal LLM $f$ parameterized by $\theta$ (consisting of two parts, $f_{vision}$ and $f_{text}$ parameterized by $\theta_{vision}$ and $\theta_{text}$) that map the input $i_e$ and $x_e$ to the prediction to $y_o$, where $i_e$ refer to the editing image input, $x_e$ refer to the editing text prompt input and $y_o$ denote as the origin output. We denote $\mathcal{M}$ as a symbolic representation for a par-

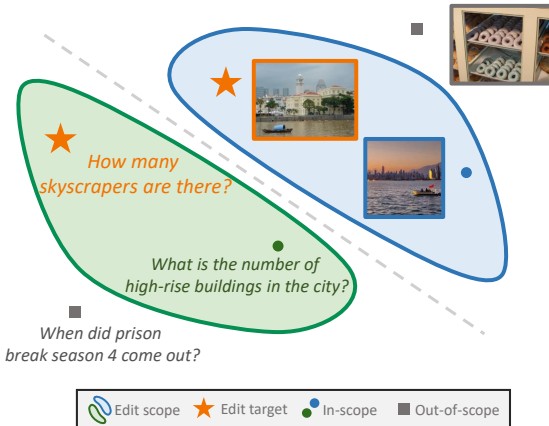

Figure 3: Taking the text modality as an example, **Edit target** and its generalization pertain to *in-scope*, which involves querying the quantity of skyscrapers in a given image, while the *out-of-scope* refers to inquiries about the publication date. In-scope inputs require editing, whereas out-of-scope inputs remain unchanged.

| TASK | Train | Test | L-Locality | M-Locality |
|------|-------|------|------------|------------|
| E-VQA | 6,346 | 2,093 | 4,289 | 5,046 |
| E-IC | 2,849 | 1,000 | 4,289 | 5,046 |

Table 1: The statistic of datasets for the E-VQA and E-IC sub-tasks. L-Locality and M-Locality are the test sets for knowledge locality to evaluate the rest of the knowledge in multimodal models when successfully updating specific facts.

ticular metric, with subscripts indicating specific metrics and superscripts representing variations in edit data. We prepare the editing datasets stated in §3.2.1, which present as $\mathcal{D}_{\text{edit}}$. Inspired by Yao et al. (2023), we introduce a series of multimodal model editing metrics.

**Reliability.** Editing reliability is needed to change prediction from $y_o$ to $y_e$. Intuitively, what we need is an updated $\theta_e$ with $f(i_e, x_e; \theta_e) = y_e$. To measure the reliability, we use the editing accuracy, as described by the following:

$$\mathcal{M}_{rel} = \mathbb{E}_{(i_e,x_e,y_e)\sim\mathcal{D}_{\text{edit}}} \left[ \mathbb{1}_{f(i_e,x_e;\theta_e(i_e,x_e,y_e))=y_e} \right] \tag{1}$$

where $\theta_e$ refers to the edited parameters.

**Locality.** To maintain the model's stability, minimizing the unintended side effects of editing on the model's broader knowledge base is imperative. In pursuit of this objective, we introduce two metrics: $\mathcal{M}_{loc}^{Text}$ (**T-Locality**) and $\mathcal{M}_{loc}^{Img}$ (**M-Locality**), both of which are designed to preserve the model's stability during the editing process. Given that the knowledge in the multimodal language model is inherited from LLMs, safeguarding this knowledge is paramount. With this aim in mind, we set aside the model's visual discrimination module and instead employ rudimentary question-and-answer datasets $\mathcal{D}_{\text{loc-t}}$ as we stated in §3.2.2. We define the question as $x$ and the answer as $y$, as below:

$$\mathcal{M}_{loc}^{Text} = \mathbb{E}_{\substack{(i_e,x_e,y_e)\sim\mathcal{D}_{\text{edit}} \\ (x,y)\sim\mathcal{D}_{\text{loc-t}}}} \left[ \mathbb{1}_{f(x;\theta_e(i_e,x_e,y_e))=f(x,\theta)} \right] \tag{2}$$

The vision encoder serves a critical function in the multimodal language model, transforming images into vector representations for co-encoding alongside natural language text. Consequently, we must take into account the potential ramifications of any modifications to this module. We construct the dataset denoted as $\mathcal{D}_{\text{loc-v}}$ for test $\mathcal{M}_{loc}^{Img}$, and calculate as delineated below:

$$\mathcal{M}_{loc}^{Img} = \mathbb{E}_{(i_v,x_v,y_v)\sim\mathcal{D}_{\text{loc-v}}} \left[ \mathbb{1}_{f(i_v,x_v;\theta_e)=f(i_v,x_v;\theta)} \right] \tag{3}$$

where $(i_v, x_v, y_v)$ is the out-of-scope data, and $\theta_e$ denote the parameter updated by edit data $(i_e, x_e, y_e)$.

**Generality.** Throughout the editing process, it is not adequate to merely amend individual erroneous inputs. The revised model should also retain the capacity for generalization and consistently produce congruent outputs for equivalent inputs (e.g., rephrased sentences), as shown in Figure 3. While previous unimodal model editing tasks only required consideration of the rephrased text, multimodal scenarios necessitate the generalization of images as well. To address this, we introduce two generalization considerations: $\mathcal{M}_{gen}^{Text}$ (**T-Generality**) and $\mathcal{M}_{gen}^{Img}$ (**M-Generality**), which are expressed as follows:

$$\mathcal{M}_{gen}^{Text} = \mathbb{E}_{(x_r)\sim\mathcal{N}(x_e)} \left[ \mathbb{1}_{f(i_e,x_r;\theta_e)=f(i_e,x_e;\theta_e)} \right] \tag{4}$$

$$\mathcal{M}_{gen}^{Img} = \mathbb{E}_{(i_r)\sim\mathcal{N}(i_e)} \left[ \mathbb{1}_{f(i_r,x_e;\theta_e)=f(i_e,x_e;\theta_e)} \right] \tag{5}$$

where $i_r$ presents the rephrased image, $x_r$ refers to the rephrased text prompt, and $\mathcal{N}(x)$ denotes to in-scope objects of $x$.

### 3.2 Datasets

The dataset **MMEdit** we constructed mainly contains two subtasks: Editing VQA (*E-VQA*) and Editing Image Captioning (*E-IC*).

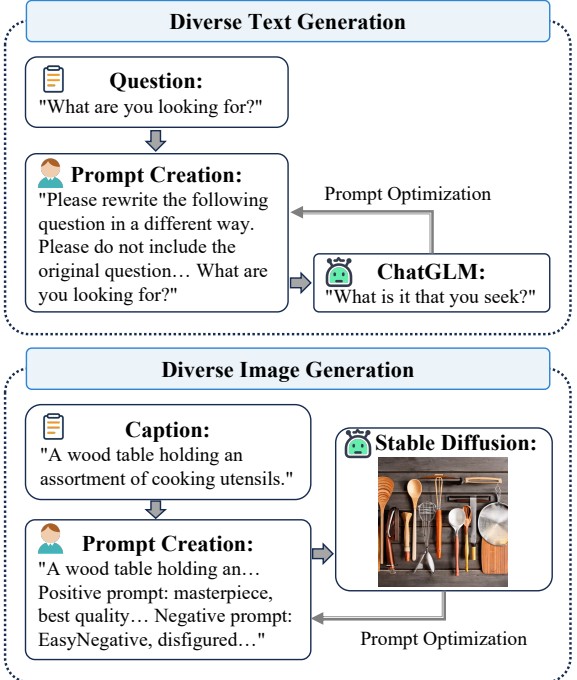

Figure 4: Generality dataset construction process.

### 3.2.1 Reliability Dataset Construction

To benchmark our experiments, we selected two common multimodal tasks: Visual Question Answering (VQA) (Antol et al., 2015) and Image Captioning (Herdade et al., 2019). VQA is to devise algorithms that can not only comprehend the visual content within an image, but also understand the natural language used to inquire about that image, and subsequently generate precise answers to those queries. Image Captioning is to devise algorithms capable of comprehending the visual content of an image, subsequently generating a coherent and precise description of the image in natural language. In this study, we opt for BLIP-2 OPT. Our foundational edit data originates from suboptimal entries across two eval datasets, namely, VQAv2 (Goyal et al., 2017) and COCO Caption (Chen et al., 2015).

Besides the foundational edit data, utilizing additional data is crucial. This data not only aids the editing process but also validates the efficacy of the changes, assessing model edits for both stability and generality.

### 3.2.2 Locality Dataset Construction

We must deliberate on the effects of editing on the language function within a multimodal model, analogous to how we evaluate various cognitive regions of an individual's brain post-surgery.

**Textual Locality Dataset.** To evaluate the stability of the language model, we leverage the NQ dataset (Kwiatkowski et al., 2019), previously used in MEND, as a benchmark for the stability of the LLM component within the model. We specifically use the model's output pre and post-editing to construct a KL scatter plot, facilitating constraints on the model's edits. Additionally, we calculate the proportion of instances maintaining a top-1 status, further quantifying the model's stability.

**MultiModal Locality Dataset.** Similarly, it's crucial to verify the impact of editing on the visual module. Hence, we utilize a straightforward dataset OK-VQA (Marino et al., 2019) in the realm of multimodality, serving as a measure of the locality for the multimodal visual module. Once again, we update the KL dispersion constraint using logits both before and after the editing process.

### 3.2.3 Generality Dataset Construction

We propose two forms of generality within a multimodal model. The overall process of generality dataset construction is shown in Figure 4.

**Textual Generality Dataset.** To be noted, LLMs exhibit robust conversational and powerfull problem-solving capabilities, which enables us to formulate task instructions, whereby we can instruct the model to produce analogous text inputs. For the E-VQA task, we utilize ChatGLM (Du et al., 2022; Zeng et al., 2022) to generate similar queries. However, for the E-IC task, due to the succinctness and relative straightforwardness of the prompts, the quality of the model's generated output is not satisfactory. Therefore, we employ a manually written template with 20 prompts to replace the original ones randomly.

**Visual Generality Dataset.** The diffusion model (Ho et al., 2020) has garnered significant success in the realm of image generation in recent years. Surpassing the original state-of-the-art model: Generative Adversarial Networks (GAN) models (Goodfellow et al., 2014). The diffusion model has excelled in numerous image-generation tasks and has shown commendable performance across various application domains. Stable Diffusion (Rombach et al., 2022) is a latent text-to-image diffusion model capable of generating photo-realistic images given text input. We utilize Stable Diffusion 2.1 for generating reinterpreted images. This dataset, drawing upon caption descriptions from the COCO dataset,

is leveraged to evaluate the model's capability for image generalization.

### 3.3 Multimodal Language Models

**BLIP-2 OPT.** BLIP-2 (Li et al., 2023b) is a generic and efficient pre-training strategy that boot-straps vision-language pre-training from off-the-shelf frozen pre-trained image encoders and frozen large language models. The model utilizes a lightweight Quering Transformer to bridge the gap between vision modality and text modality and achieves state-of-the-art performance on various vision-language tasks. We select the BLIP-2 OPT as our basic edit model, which utilizes the ViT-L in the vision block, and select the unsupervised-trained OPT model for decoder-based LLM.

**MiniGPT-4.** MiniGPT-4 (Zhu et al., 2023) is a potent vision-language model akin to BLIP-2, leveraging a frozen visual encoder in tandem with the frozen Vicuna (Chiang et al., 2023). Vicuna, built upon LLaMA, is reported to achieve 90% of Chat-GPT's performance based on GPT-4's evaluation criteria. MiniGPT-4 adds a single projection layer to align the encoded visual features with the Vicuna language model. And MiniGPT-4 employs the same pre-trained vision component of BLIP-2 that consists of a Vit-G/14 from EVA-CLIP (Sun et al., 2023) and a Q-Former.

### 3.4 Baselines

**Finetune.** Fine-tuning has emerged as a widely employed strategy for adapting pre-trained language models to specific tasks or domains (Cortes et al., 2015). In our exploration, we delve into two distinct fine-tuning methodologies: one focusing on the **last layer** of the language model. Take the BLIP-2 OPT model as an example, we finetune the 31st decoder layer of OPT model. The other targets the **vision block** within the multimodal language model, specifically, we finetune the Q-former model to overfit the editing dataset.

**MEND.** Model Editor Networks with Gradient Decomposition (Mitchell et al., 2022a) conducts efficient local edits to language models with a single input-output pair. Essentially, MEND learns to transform the gradient of fine-tuned LLMs, which utilizes a low-rank decomposition of gradients.

**Knowledge Editor.** KE (Cao et al., 2021) is a method that can edit wrong knowledge in language models without re-training the whole model. KE utilizes a hyper network (a bidirectional-LSTM) with constrained optimization, which is used to predict the weight update during inference.

**SERAC.** SERAC (Mitchell et al., 2022b) introduces a memory-based model editing approach, which leverages an explicit memory system to cache edits. This memory is subsequently used to adjust the output of the base model during inference. The system utilizes a small auxiliary *scope classifier* alongside *counterfactual model*. The role of the scope classifier is to ascertain whether the input is within the ambit of the memory cache. Should the input be found within this scope, it is combined with the most relevant cache item and input into the counterfactual model for prediction.

**In-Context Knowledge Editing.** In-Context Knowledge Editing (IKE) (Zheng et al., 2023) constructs $k$ demonstrations $C = \{c_1, \ldots, c_k\}$, following the approach outlined in Liu et al. (2022). This method employs an unsupervised retriever based on cosine similarity to fetch demonstrations from the training set prior to injecting fact $f = (x^*, y^*)$ into Language Models. The $x^*$ is the prompt to probe the factual knowledge in models (e.g., `The president of the US is`), and $y^*$ will be the editing target `Joe Biden`. The ranking of in-context demonstrations also hinges on cosine similarity: $cos(c_1, f) < cos(c_2, f) < \cdots < cos(c_k, f)$. where $c_1, \ldots, c_k$ are sequentially arranged in the context from left to right. Demonstrations $C$ can be viewed as an externally augmented knowledge base, primarily designed to guide the generation within LMs. Its ultimate objective is to maximize $\mathcal{P}(y \mid x, f, C)$ when the prompt $x$ falls within the editing scope of the target prompt $x^*$.

## 4 Experiments

### 4.1 Results

In this part, we present a comparative analysis of multiple editing methods on **MMEdit**. The results of these comparisons are displayed in Table 2. After this, we delve into a tripartite evaluation of the experimental results, including three aspects of **Reliability**, **Locality**, and **Generality**. Furthermore, we analyze Locality and Generality through text and visual modalities and provide several editing cases in Figure 6.

**Reliability.** From the results, all model editing methods outperform the base methods in Reliabil-

| | Method | Editing VQA | | | | Editing Image Caption | | | |
|---|---|---|---|---|---|---|---|---|---|
| | | Reliability ↑ | T-Generality ↑ | T-Locality ↑ | M-Locality ↑ | Reliability ↑ | T-Generality ↑ | T-Locality ↑ | M-Locality ↑ |
| **BLIP-2 OPT** | | | | | | | | | Size: 3.8B |
| Base Methods | Base Model | 0.00 | 0.00 | 100.0 | 100.0 | 0.00 | 0.00 | 100.0 | 100.0 |
| | FT (vision block) | 56.28 | 29.88 | 100.0 | 11.32 | 0.08 | 0.00 | 100.0 | 7.31 |
| | FT (last layer) | 58.70 | 15.33 | 78.86 | 2.86 | 0.24 | 0.10 | 67.67 | 3.91 |
| Model Editing | Knowledge Editor | 67.80 | 63.00 | 97.32 | 45.89 | 69.00 | 62.80 | 96.21 | 45.55 |
| | In-Context Editing | 99.95 | 91.59 | 13.16 | 1.88 | 96.70 | 78.20 | 13.36 | 2.17 |
| | SERAC | 91.20 | 91.40 | 100.0 | 0.33 | 94.40 | 96.00 | 100.0 | 0.47 |
| | MEND | 92.60 | 90.80 | 96.07 | 65.15 | 65.00 | 38.00 | 92.67 | 55.72 |
| **MiniGPT-4** | | | | | | | | | Size: 7.3B |
| Base Methods | Base Model | 0.00 | 0.00 | 100.0 | 100.0 | 0.00 | 0.00 | 100.0 | 100.0 |
| | FT (vision block) | 39.58 | 0.98 | 100.0 | 3.96 | 0.63 | 0.00 | 100.0 | 5.13 |
| | FT (last layer) | 39.57 | 0.58 | 72.01 | 16.42 | 2.75 | 0.00 | 35.52 | 9.28 |
| Model Editing | Knowledge Editor | 87.77 | 86.62 | 97.15 | 55.77 | 35.10 | 24.20 | 96.78 | 52.22 |
| | In-Context Editing | 71.72 | 40.23 | 13.46 | 2.00 | 68.60 | 59.80 | 12.51 | 2.96 |
| | SERAC | 87.20 | 84.60 | 100.0 | 0.33 | 40.20 | 36.60 | 100.0 | 0.97 |
| | MEND | 95.51 | 95.27 | 98.73 | 71.33 | 87.10 | 84.10 | 98.34 | 59.53 |

Table 2: Main results on the **MMEdit**. **T-Locality, M-Locality** refer to the textual and multimodal stability. **T-Generality** represents textual generality. **Reliability** denotes the accuracy of successful editing.

ity. Particularly, IKE and SERAC, methodologies leveraging external memory for editing, exhibit commendable performance in multimodal language models. We observe that the fine-tuning method demonstrates poorer performance than the model editing method. Note that merely fine-tuning the parameters of the LLM or the modal fusion block does not adequately capture the characteristics of the multimodal data. We analyze the reasons as

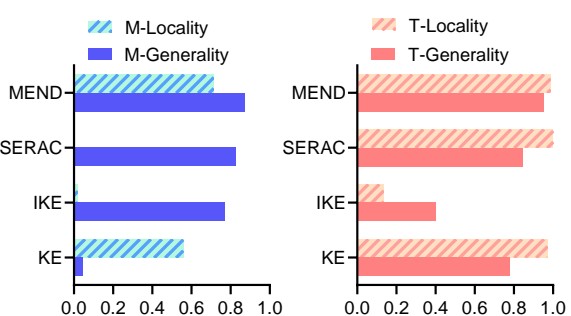

Figure 5: Generality of different editing methods.

follows: the data used for fine-tuning differs significantly from the original model, such as the Q-former and OPT model, which need to collaborate effectively. Simply fine-tuning one of these modules may not capture the task-specific characteristics accurately. On the other hand, fine-tuning all modules incurs a significant resource overhead. Moreover, based on our experimental results, we observe that fine-tuning can lead to substantial changes in the original model, often resulting in the loss of other knowledge, particularly evident in multimodal datasets.

**Locality.** Several traditional editing methods remain applicable in multimodal editing, proving valuable for effectively modifying the knowledge within the model and rectifying its outputs. However, IKE and SERAC, despite their superior performance in Reliability, exhibit poor performance on the M-Locality due to their lack of constraints on it, indicating that although these external memory-based editing techniques undoubtedly succeed in fixing the outputs, their efficacy in stabilizing internal knowledge within the models leaves room for improvement. As for T-Locality, the majority of Model Editing methods obtain good performance, with IKE once again falling short. The underlying reason is that the other three approaches impose constraints on T-Locality, whereas IKE, as an In-Context Learning method, lacks a robust constraint mechanism, resulting in subpar performance.

**Generality.** We undertake a comparative exploration of various methods' text and image generalization capabilities with MiniGPT-4 in E-VQA. Note that KE tends to display a lesser degree of image generalization, predominantly due to its inherent consideration of M-Locality during the training phase. Consequently, the image generalization efficiency of meta-learning methods tends to fall short when compared to memory-based methods. On the other hand, the superior image generalization capability exhibited by memory-based methods is achieved at the cost of compromising M-Locality, resulting in significantly lower levels of M-Locality. Through our evaluation of diverse editing methods,

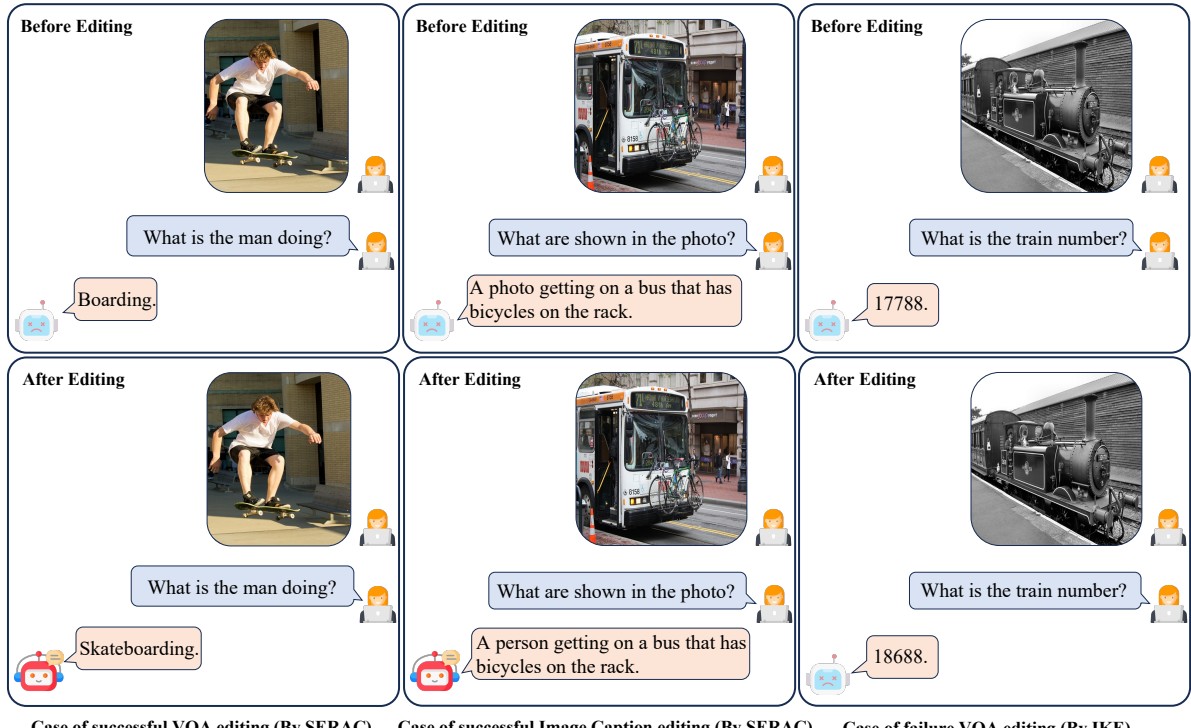

Figure 6: Cases of multimodal model editing. **Top**: The output before editing. **Bottom**: The output after editing.

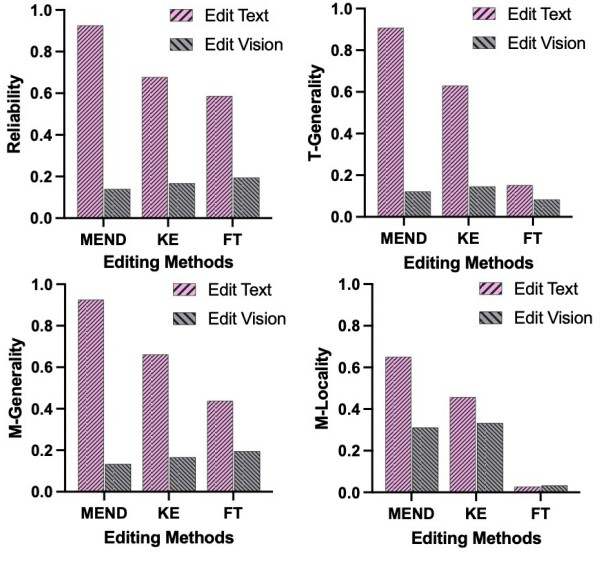

Figure 7: Results of editing different components.

we recurrently identify that image generalization performance tends to be less robust than text generalization.

## 4.2 Editing Different Component

We further analyze the variations in editing different regions of the multimodal model. In contrast to editing single-modal models, due to the complexity and diversity of multimodal models, we can try to

edit more modules and analyze their impact on visual and textual knowledge. The results are shown in Figure 7. For the BLIP-2 OPT model, we investigate the distinctions in editing the Q-former and OPT on the VQA dataset. Regarding the MiniGPT-4 model, we mainly focus on the distinctions in editing the last few layers of the *llama_proj* and Vicuna models. The selected editing approaches for analysis are MEND, KE, and FT, which enable us to specify the editing area.

The results highlight that editing the vision module is more challenging than editing the language module (also see the failure editing in Figure 6). We argue that this difficulty may be attributed to the model's architecture. Editing the last layer of the LLM allows for direct modification of the output, while modifying the vision module only affects the input to the LLM, resulting in relatively less impact on the model. Concretely, various modalities reside in distinct spaces, which implies that the factual knowledge may be stored in separate parameters within the model. Considering that the LLMs possess a large number of parameters, this aspect becomes even more critical for multimodal models. Thus editing the language model can lead to significant performance improvements. Notably, the visual module in the model plays a crucial role in image comprehension, thereby suggesting that

future work needs to **consider information from different modalities simultaneously**.

## 5 Conclusion

In this paper, we introduce multimodal model editing, with a new benchmark **MMEdit**. Empirically, we analyze the effectiveness of various model editing baselines and explore their impact on different components (e.g., visual and text).

## Acknowledgment

We would like to express gratitude to the anonymous reviewers for their kind comments. This work was supported by the National Natural Science Foundation of China (No.62206246), Zhejiang Provincial Natural Science Foundation of China (No. LGG22F030011), Ningbo Natural Science Foundation (2021J190), Yongjiang Talent Introduction Programme (2021A-156-G), Zhejiang Provincial Science and Technology Plan Project (2023C01120), CCF-Tencent Rhino-Bird Open Research Fund, and Information Technology Center and State Key Lab of CAD&CG, Zhejiang University.

## 6 Limitations

**Models.** We only edit several basic multimodal LLMs, leaving many others behind. Besides, due to the resource limitation, the number of parameters for the multimodal LLMs we edit is below 10B, and we cannot afford to edit LLMs with a larger number of parameters such as the 65B LLaMA Adapter V2 (Gao et al., 2023).

**Efficient Vision Editing.** In this paper, our analysis has been primarily focused on comparing the varied effects of existing editing methods across modules of different modalities. However, the results are not satisfactory. Moving forward, our primary objective is to explore how to efficiently and accurately edit information across other modalities. This includes investigating techniques such as co-editing between different modalities by pinpointing the knowledge within the multimodal model and identifying the content requiring modification.

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

# A  Appendix

**Our code is available in the supplementary materials for reproducibility**.

| Hyper-Parameters | MaxIter | Edit Num | Optimizer | LR |
|---|---|---|---|---|
| $D_{BLIP2}^{E-VQA}$ | 40000 | 1 | ASGD | 1e-5 |
| $D_{BLIP2}^{E-IC}$ | 40000 | 1 | ASGD | 1e-5 |
| $D_{MiniGPT-4}^{E-VQA}$ | 40000 | 1 | ASGD | 1e-5 |
| $D_{MiniGPT-4}^{E-IC}$ | 40000 | 1 | ASGD | 1e-5 |

Table 3: FT-vision hyper-parameters

| Hyper-Parameters | MaxIter | Edit Num | Optimizer | LR |
|---|---|---|---|---|
| $D_{BLIP2}^{E-VQA}$ | 20000 | 1 | ASGD | 1e-5 |
| $D_{BLIP2}^{E-IC}$ | 20000 | 1 | ASGD | 1e-5 |
| $D_{MiniGPT-4}^{E-VQA}$ | 20000 | 1 | ASGD | 1e-5 |
| $D_{MiniGPT-4}^{E-IC}$ | 20000 | 1 | ASGD | 1e-5 |

Table 4: FT-last-layer hyper-parameters

In this section, we describe the implementation of our experiments in detail, including the training procedures, backbone model, and hyperparameters for each dataset.

| Hyper-Parameters | MaxIter | Edit Num | Optimizer | LR |
|---|---|---|---|---|
| $D_{BLIP2}^{E-VQA}$ | 20,000 | 1 | Adam | 1e-5 |
| $D_{BLIP2}^{E-IC}$ | 20,000 | 1 | Adam | 1e-5 |
| $D_{MiniGPT-4}^{E-VQA}$ | 25,000 | 1 | AdamW | 5e-4 |
| $D_{MiniGPT-4}^{E-IC}$ | 35,000 | 1 | AdamW | 5e-4 |

Table 5: KE hyper-parameters

| Hyper-Parameters | MaxIter | Edit Num | Optimizer | LR |
|---|---|---|---|---|
| $D_{BLIP2}^{E-VQA}$ | 15,000 | 1 | Adam | 1e-5 |
| $D_{BLIP2}^{E-IC}$ | 15,000 | 1 | Adam | 1e-5 |
| $D_{MiniGPT-4}^{E-VQA}$ | 20,000 | 1 | Adam | 1e-5 |
| $D_{MiniGPT-4}^{E-IC}$ | 30,000 | 1 | Adam | 1e-5 |

Table 6: SERAC hyper-parameters

| Hyper-Parameters | MaxIter | Edit Num | Optimizer | LR |
|---|---|---|---|---|
| $D_{BLIP2}^{E-VQA}$ | 20,000 | 1 | Adam | 1e-6 |
| $D_{BLIP2}^{E-IC}$ | 20,000 | 1 | Adam | 1e-6 |
| $D_{MiniGPT-4}^{E-VQA}$ | 20,000 | 1 | Adam | 1e-6 |
| $D_{MiniGPT-4}^{E-IC}$ | 20,000 | 1 | Adam | 1e-6 |

Table 7: MEND hyper-parameters