# OpenReview forum: "Can We Edit Multimodal Large Language Models?"
_EMNLP/2023/Conference — EMNLP 2023 Main_

### Official Review · Reviewer_hZHi · 2023-07-30

**Soundness:** 4

**Excitement:**

3: Ambivalent: It has merits (e.g., it reports state-of-the-art results, the idea is nice), but there are key weaknesses (e.g., it describes incremental work), and it can significantly benefit from another round of revision. However, I won't object to accepting it if my co-reviewers champion it.

**Paper Topic And Main Contributions:**

This paper focuses on the task of editing multimodal large language models (LLMs), which is  more challenging compared to editing single-modal LLMs. To facilitate research in this area, the authors  construct a benchmark called MMEdit and propose innovative metrics for evaluating multimodal LLM editing. They  conduct comprehensive experiments with various editing baselines and analyze the impact of editing different components.  The results indicate that while it is possible to edit multimodal LLMs to some extent,  the effectiveness is still limited.

**Reasons To Accept:**

1) The paper introduces a new and challenging task of editing multimodal LLMs, which extends  the existing knowledge editing methodologies for single-modal LLMs.
2) The authors provide a detailed description of the MMEdit benchmark, which includes metrics for evaluation. This benchmark can facilitate future research in editing multimodal LLMs.
3) The comprehensive experiments and analysis of the impact of editing different components provide valuable  insights into the editing task.

**Reasons To Reject:**

1) The description in the section of dataset construction is not detailed enough and insufficient to understand how the data was constructed.
2) The conclusion of the experimental section does not feel very innovative.

**Reproducibility:**

4: Could mostly reproduce the results, but there may be some variation because of sample variance or minor variations in their interpretation of the protocol or method.

**Reviewer Confidence:**

1: Not my area, or paper was hard for me to understand. My evaluation is just an educated guess.

---

> ### Author Rebuttal · Authors · 2023-08-29
>
> We sincerely appreciate your time in reading the paper, and our point-to-point responses to your comments are given below.
>
> > The description in the section of dataset construction is not detailed enough and insufficient to understand how the data was constructed.
>
> We regret not delving deeper into the dataset construction in Section 3 because of space limitations. We'll endeavor to provide more comprehensive details on dataset construction and aim to make our data publicly accessible, accompanied by more illustrative samples, in the near future.
>
> > The conclusion of the experimental section does not feel very innovative.
>
> We regret not delving deeper into the conclusion. Space limitations led us to provide a concise summary, but we've detailed most results and rationales in Section 4. We'll strive to enhance this section in upcoming revisions.

---

### Official Review · Reviewer_ysoz · 2023-08-11

**Typos Grammar Style And Presentation Improvements:** 1. The link for stable diffusion 2.1 …
**Soundness:** 4

**Excitement:**

3: Ambivalent: It has merits (e.g., it reports state-of-the-art results, the idea is nice), but there are key weaknesses (e.g., it describes incremental work), and it can significantly benefit from another round of revision. However, I won't object to accepting it if my co-reviewers champion it.

**Paper Topic And Main Contributions:**

The manuscript proposed MMEdit (Multimodal Model Editing) and presented its evaluations on two tasks: Editing VQA (Visual Question-Answering) and Editing Image captioning. The work highlights that current editing works focus on the textual LLMs, and the authors extend the editing to multimodal LLMs and show the observations that current editing works fall short on the multimodal LLMs and observe a potential gap in the editing techniques by proposing MMedit.

The work illustrates the results of two models (BLIP-2 OPT and MiniGPT-4) in editing VQA and Image caption task. The base methods (FT) and Editing (KE, ICE, SERAC, and MEND) results are highlighted with capability on the evaluation of reliability, generality, and locality.

**Questions For The Authors:**

Question A: What is the editing prompt in Figure 6?
Question B: There are other ME techniques that could also be experimented with, say, CaliNet, T-Patcher, KN, ENN, CuQA, SLAG, FRUIT, Prompt-editing, FactTracing, RepairNN, PAINT, Enatailer, GRACE, Fact, and similar. The current set of MEND, SERAC, and KE belongs to the hypernetwork-based approach. But locate-then-edit or parameter-preserving approaches might show better results. One suggestion is to show comparisons with the existing ME techniques. A discussion on the problems, challenges, and possibilities of the existing techniques would help in highlighting the limitations of other editing techniques.
Question C: What would be the hyperparameters or experiment settings? Since the appendix is missing, the exact reproducibility is challenging. It is really good to see that the code and datasets will be made public. The proposed work and problem approach will definitely help the NLP community.
Question D: In Table 2, what are the reasons for BLIP2-OPT to perform worst on reliability and T-Generability, but better on T-Locality and M-locality? The insights and observations from the table are not clear.

**Reasons To Accept:**

1. The presentation of the problem is well understood, and the works highlight the burning need for editing in multimodal LLMs, which definitely adds to aid the NLP community.
2. The manuscript covers the essential and necessary problem of editing. The insights are definitely helpful for the NLP community.
3. The set of ME techniques in Table 2 covers standard editing techniques from two paradigms (Hypernetwork-based and parameter-preserving). It shows an insightful observation over two different datasets and four metrics.

**Reasons To Reject:**

The reasons for the rejection are as follows:
1. The proposed problem in the manuscript is no doubt a great contribution, but the insights from the editing techniques and observations are not primarily clear.
2. An analysis of the performance degradation, for instance, after editing with 10 instances, is not available in the manuscript.
3. An appendix with the implementation details can help with reproducibility. Additionally, the wall clock time to edit one instance and GPU consumption details are an add-on to show the analysis could also help in the insights.
4. The manuscript covers the discussion on a range of editing techniques but shows the comparison of editing on four techniques. The discussion and an analysis of the locate-then-edit approach for multimodal LLMs can show better insights with comparable observations.

**Reproducibility:**

4: Could mostly reproduce the results, but there may be some variation because of sample variance or minor variations in their interpretation of the protocol or method.

**Reviewer Confidence:**

4: Quite sure. I tried to check the important points carefully. It's unlikely, though conceivable, that I missed something that should affect my ratings.

---

> ### Author Rebuttal · Authors · 2023-08-29
>
> We appreciate your comprehensive feedback. Please find detailed responses in the following.
>
> > Question A) What is the editing prompt in Figure 6?
>
> IKE[1] utilizes demonstrations to guide LMs to edit knowledge facts by in-context learning (ICL). In alignment with the original paper, we choose k-NN examples from the training corpus. The demonstrations are encoded by *all-MiniLM-L6-v2*. Following the default setting, we set k to 32. Due to the extensive length of the prompts in IKE, we only enumerate two representative examples here.
> The examples for demonstration and target are as follows:
> | Type | Text |
> | -------- | -------- |
> | Demonstration | New Fact: How many balls are on the field? 1\n Q: How many balls are on the field? A: 1 |
> | Target | New fact: What is the train number? 32678\n Q: What is the train number? A: |
>
> > Question B) There are other ME techniques that could also be experimented with, say, CaliNet, T-Patcher, KN, ENN, CuQA, SLAG, FRUIT, Prompt-editing, FactTracing, RepairNN, PAINT, Enatailer, GRACE, Fact, and similar. The current set of MEND, SERAC, and KE belongs to the hypernetwork-based approach. But locate-then-edit or parameter-preserving approaches might show better results. One suggestion is to show comparisons with the existing ME techniques. A discussion on the problems, challenges, and possibilities of the existing techniques would help in highlighting the limitations of other editing techniques.
>
> We acknowledge the breadth and significance of the ME techniques you've pointed out. Indeed, the **locate-then-edit** approach, exemplified by methods such as ROME and MEMIT, focuses on updating facts in the format (subject s, relation r, object o). This method requires pinpointing the location of the last token of the subject within the input before proceeding to the edit. However, in our task, multimodal datasets lack a specific subject to locate, rendering this approach inapplicable.
> Regarding the parameter-preserving approach, the survey[2] indicates that CaliNet exhibits suboptimal performance in single-modality. Consequently, we did not select it as a baseline.
>
> > Question C) What would be the hyperparameters or experiment settings? Since the appendix is missing, the exact reproducibility is challenging. It is really good to see that the code and datasets will be made public. The proposed work and problem approach will definitely help the NLP community.
>
> We appreciate your interest in the experiment details. The hyperparameters and experimental settings can be found in the **Supplementary Materials**. In our revised manuscript, we will also incorporate these details in the appendix for clarity. Below, we provide the hyperparameters of MEND.
> | Hyper-Parameters | MaxIter | Edit Num | Optimizer  | LR |
> | :----- | ----- | ----- | ----- | ----- |
> | $D_{BLIP2}^{E-VQA}$ | 20,000 | 1 | Adam | 1e-6 |
> | $D_{BLIP2}^{E-IC}$ | 20,000 | 1 | Adam | 1e-6 |
> | $D_{MiniGPT-4}^{E-VQA} $| 20,000 | 1 | Adam | 1e-6 |
> | $D_{MiniGPT-4}^{E-IC} $| 20,000 | 1 | Adam | 1e-6 |
> Furthermore, we will release our code and datasets publicly, aligning with our dedication to contributing to the NLP community.
>
> > Question D) In Table 2, what are the reasons for BLIP2-OPT to perform worst on reliability and T-Generability, but better on T-Locality and M-locality? The insights and observations from the table are not clear.
>
> Thank you for pointing out the observations from Table 2. This behavior of BLIP2-OPT can be attributed to the characteristics of certain meta-learning methods. Methods like KE and MEND consider both accuracy and locality in their training objectives. This approach can sometimes lead to trade-offs: doing well in one metric might impact another. When these methods focus on improving locality, they might do better on T-locality and M-locality but might not perform as well on reliability and T-Generability.
>
> > Typos Grammar Style And Presentation Improvements
>
> For typos and presentation: We appreciate your feedback on improving our paper. We have revisited those sections and have made changes to enhance the overall presentation.
>
> [1] [Can We Edit Factual Knowledge by In-Context Learning?](https://arxiv.org/abs/2305.12740)
>
> [2] [Editing Large Language Models: Problems, Methods, and Opportunities.](https://arxiv.org/abs/2305.13172)

---

### Official Review · Reviewer_sCgZ · 2023-08-11

**Soundness:** 4

**Excitement:**

3: Ambivalent: It has merits (e.g., it reports state-of-the-art results, the idea is nice), but there are key weaknesses (e.g., it describes incremental work), and it can significantly benefit from another round of revision. However, I won't object to accepting it if my co-reviewers champion it.

**Missing References:**

N/A

**Paper Topic And Main Contributions:**

The paper presents MMEdit, a benchmark for evaluating editing of multimodal LLMs. The paper describes the task and three axes of evaluation: whether an editing technique successfully corrects the incorrect outputs (Reliability), whether it handles the edit examples and related examples in a consistent way (Generality), and whether it keeps outputs on unrelated inputs the same (Locality); the latter two are measured separately for the image and text modalities. A few editing techniques and baselines are compared using two backbone multimodal LLMs.

**Questions For The Authors:**

Question A) If I understand the metric equations correctly, the expectations are over cases where you use a single edit example to update the model; if this is the case, is there a way to test a technique's performance when a larger set of edits are made? I could imagine the different techniques having different behaviors as the number of edits increases.

Question B) In line 264, what does "in-scope objects of x" mean? Specifically, what are the objects, and how do you determine which are in-scope?

Question C) For measuring Locality, is there any guarantee that the examples in D_loc-t and D_loc-v are truly unrelated to the edit examples? For example, if you edit the model based on a VQA example from one dataset, if there are similar examples in the other VQA dataset used as D_loc-v then seeing a change there might actually be desirable.

Question D) In the paragraph starting on line 288, what is the "supplementary data"? It seems like the description of it is missing; if it is present then this paragraph should be moved or re-written to make it clear what is being referred to.

Question E) Why are the results split between Table 2 and Figure 5? I would prefer having a complete view of all main results in a single table or figure.

Question F) Is the discussion of SERAC starting at line 483 correct? The model's image generalization seems fine, and I don't think it has "inherent consideration of M-Locality during the training phase" because it has terrible performance in M-Locality. Is this just a typo and you meant some system other than SERAC, like KE?

**Reasons To Accept:**

 - The paper's application of model editing to multimodal LLMs appears to be novel.
 - The authors have provided and will release code and data for their benchmark, allowing the community to use and iterate on it.
 - The paper proposes a way to extend existing model-editing metric concepts to the multimodal setting.

**Reasons To Reject:**

 - There are a few details that appear to be missing from the paper; see Questions B and D.

**Reproducibility:**

4: Could mostly reproduce the results, but there may be some variation because of sample variance or minor variations in their interpretation of the protocol or method.

**Reviewer Confidence:**

3: Pretty sure, but there's a chance I missed something. Although I have a good feel for this area in general, I did not carefully check the paper's details, e.g., the math, experimental design, or novelty.

**Typos Grammar Style And Presentation Improvements:**

 - The equations for text locality (line 234) and image locality (line 243) are inconsistent: the first includes the expectation over elements from D_edit but the second does not.
 - Figures 3 and 4  and Table 1 are not referenced in the body of the text; I recommend referencing all tables/figures in the text somewhere, or you run the risk of the reader not looking at them.
- Line 412-413: "construct" -> "constructs"
- Line 428: "Experiment" -> "Experiments"
- Line 538: "Limitation" -> "Limitations"
- Line 417: x* and y* appear to be undefined.

---

> ### Author Rebuttal · Authors · 2023-08-29
>
> Thanks for reviewing our paper and the valuable feedback. In addition to the general updates, we address your concerns here.
> > Question A) If I understand the metric equations correctly, the expectations are over cases where you use a single edit example to update the model; if this is the case, is there a way to test a technique's performance when a larger set of edits are made? I could imagine the different techniques having different behaviors as the number of edits increases.
>
> Experiments exploring various numbers of edits are crucial in model editing to understand the impact of edit quantity. We omitted this experiment from our paper for two reasons: First, previous research in model editing (MEMIT[1], Survey[2]) has indicated a diminishing effect as the number of edits increases. Secondly, our limited computational resources prevented us from executing the n-edits experiment. We revisit the n-edits experiment yet regret our inability to compare against all baseline methods due to constraints in time and resources. We evaluated MEND's efficacy in editing the MiniGPT-4 model on E-VQA using varying edit counts, specifically {1,2,4,8}, with the results detailed below:
> | Edit Num | Reliability | T-Generality | T-Locality | M-Locality |
> | -------- | -------- | ----------- | --------- | -------- |
> | 1 | 95.51 | 95.27 | 98.73 | 71.33 |
> | 2 | 94.40 | 94.80 | 97.83 | 61.18 |
> | 4 | 93.30 | 93.75 | 97.50 | 55.85 |
> | 8 | 75.87 | 72.90 | 93.00 | 40.44 |
> > Question B) In line 264, what does "in-scope objects of x" mean? Specifically, what are the objects, and how do you determine which are in-scope?
>
> I apologize for the confusion our unclear definitions may have caused. We would like to clarify that $\mathcal{N}(x)$ represents the set of synonyms: in text, it signifies semantic similarities, and in images, it signifies analogous images. For the E-VQA task, each $x_e$ corresponds to one entry as $x_r$ from similar statements. For the E-IC task, we use a manually crafted template to randomly select an entry as $x_r$. The $i_r$ for images are those akin to $i_e$, produced through stable diffusion.
>
> > Question C) For measuring Locality, is there any guarantee that the examples in D_loc-t and D_loc-v are truly unrelated to the edit examples? For example, if you edit the model based on a VQA example from one dataset, if there are similar examples in the other VQA dataset used as D_loc-v then seeing a change there might actually be desirable.
>
> We wish to emphasize that the data for the locality dataset is chosen from sources external to the training dataset and valid dataset, **ensuring no data overlap**.
>
> >Question D) In the paragraph starting on line 288, what is the "supplementary data"? It seems like the description of it is missing; if it is present then this paragraph should be moved or re-written to make it clear what is being referred to.
>
> We apologize that we didn't articulate the meaning of **supplementary data** clearly here. In this context, **supplementary data** refers to the generality and locality datasets. Our primary argument is that merely a reliable dataset for editing is insufficient; additional datasets are essential to assess the impact of editing. We've clarified this in the revised article.
>
> > Question E) Why are the results split between Table 2 and Figure 5? I would prefer having a complete view of all main results in a single table or figure.
>
> We presented the generalizability experimental results separately from the main table to offer readers a clearer and more intuitive grasp of the performance in terms of both generality and stability across different editing methods.
>
> >Question F) Is the discussion of SERAC starting at line 483 correct? The model's image generalization seems fine, and I don't think it has "inherent consideration of M-Locality during the training phase" because it has terrible performance in M-Locality. Is this just a typo and you meant some system other than SERAC, like KE?
>
> I apologize for the oversight and any inconvenience it may have caused. We've addressed this with a revision in the article.
>
> > Typos Grammar Style And Presentation Improvements
>
> We've corrected all the spelling and grammatical errors you pointed out in the article. Thank you for highlighting those issues.
>
> [1] [Mass-Editing Memory in a Transformer.](https://arxiv.org/abs/2210.07229)
>
> [2] [Editing Large Language Models: Problems, Methods, and Opportunities.](https://arxiv.org/abs/2305.13172)

---

### Meta-Review · Area_Chair_Dtyd · 2023-09-10

**Recommendation:** 4

**Metareview:**

This papers presents a benchmark aiming at facilitating the evaluation of editing knowledge in multimodal LLMs, specifically the reliability, locality, and generality evaluation metrics are the focus of this paper. Two tasks are included Image  Captioning and VQA. Data and code will be made available.
A general synthesis of the various results would help the reader to have a clear understanding of the results ans insights.

---

### Decision · Program_Chairs · 2023-10-07

**Decision:**

Accept-Main

**Comment:**

This papers presents a benchmark aiming at facilitating the evaluation of editing knowledge in multimodal LLMs, specifically the reliability, locality, and generality evaluation metrics are the focus of this paper. Two tasks are included Image  Captioning and VQA. Data and code will be made available.
A general synthesis of the various results would help the reader to have a clear understanding of the results ans insights.